# Impact of Changes in Time Left Alone on Separation-Related Behaviour in UK Pet Dogs

**DOI:** 10.3390/ani12040482

**Published:** 2022-02-15

**Authors:** Naomi D. Harvey, Robert M. Christley, Kassandra Giragosian, Rebecca Mead, Jane K. Murray, Lauren Samet, Melissa M. Upjohn, Rachel A. Casey

**Affiliations:** Canine Behaviour and Research, Dogs Trust, London EC1V 7RQ, UK; robert.christley@dogstrust.org.uk (R.M.C.); kassandra.giragosian@dogstrust.org.uk (K.G.); rebecca.mead@dogstrust.org.uk (R.M.); jane.murray@dogstrust.org.uk (J.K.M.); lauren.samet@dogstrust.org.uk (L.S.); melissa.upjohn@dogstrust.org.uk (M.M.U.); rachel.casey@dogstrust.org.uk (R.A.C.)

**Keywords:** separation-related behaviour, dog, COVID-19, clinical behaviour, risk factors, separation-anxiety, problem behaviour, SRB

## Abstract

**Simple Summary:**

Dogs can develop separation-related behaviours (SRBs), such as barking/howling, pacing, toileting or destroying household items when they are left without human company. These behaviours can be problematic for owners but are also welfare concerns for the dogs because they mean the dog is in a negative emotional state. Many dog owners have been able to spend more time at home with their pets during the COVID-19 pandemic, and there have been concerns this will increase risks of SRB when dogs are left alone for longer again. Here, we surveyed a group of dog owners in the United Kingdom (UK) twice during 2020. In the first survey, we asked owners about their dogs’ behaviour before the pandemic, in February 2020, and at the time of the first lockdown; we then surveyed them again in October 2020 when restrictions had eased. Whether dogs showed SRBs or not changed considerably over the months of the study, and whilst one in two dogs with pre-existing SRBs were no longer showing them in October 2020, one in ten dogs who had previously not shown SRBs before the pandemic were found to have developed SRBs in October. Risk of developing SRBs was found to be linked to the change in time left alone during the lockdown, with dogs whose time alone reduced the most being most at risk of developing new SRBs in October.

**Abstract:**

Separation-related behaviours (SRBs), including but not limited to vocalisation, pacing, destruction and toileting, occur in the absence of human company. As well as being problematic for the dogs’ owners, such behaviours indicate that the dogs’ emotional state is compromised. As part of the COVID-19 pandemic, time spent alone decreased considerably for many pet dogs, leading to concerns about the development of SRBs when dogs are left alone more again. The main aim of this study was to test the hypothesis that dogs whose time left alone decreased most (compared to a February 2020 pre-COVID baseline) would be at greatest risk of developing new signs of SRB when time left alone increased again. To achieve this aim, we utilised survey data gathered from dog owners between 4th May and 3rd July 2020, during the first COVID-19 ‘lockdown’ period in the United Kingdom (UK), and a follow-up survey of the same dog-owner cohort, completed when restrictions had eased between 10th October and 2nd November 2020. Individual dogs fluctuated considerably in whether they showed signs of SRB or not across the study period (*n* = 1807). Overall, the prevalence of SRB in the population decreased from 22.1% to 17.2%, as did the time dogs were left alone for between February and October 2020. However, 9.9% of dogs had developed new signs of SRB by the follow-up survey in October 2020, with dogs whose leaving hours decreased most during lockdown restrictions being at increased risk of developing SRBs. These findings have implications for our understanding of the etiology of SRB, by showing a link between changes in owner routine and SRB risk.

## 1. Introduction

Separation-related behaviours (SRBs), sometimes referred to as separation-related problems, in companion dogs commonly include destruction, vocalization, and house soiling (e.g., urination, defecation, vomiting), as well as pacing, restlessness or owner searching, panting, hypervigilance, and/or extreme passivity [1,2,3,4]. When such behaviour occurs in the absence of, or lack of access to, human company (or in some cases, specific people), they can be considered separation-related behaviours [5]. Dogs displaying SRBs are often in a compromised emotional state, with anxiety (an emotional reaction to a potential or predicted danger or uncertainty [6]) being one of the commonly associated states for dogs displaying SRBs. However, research has also suggested frustration, panic, fear, and boredom, as other negative affective states associated with different types of SRBs [1]. In addition to being a welfare concern for the dog, SRBs are often unwanted or problematic for the owner and/or neighbours, such as in the case of excessive vocalisation, which may lead to noise complaints, or in terms of house soiling and destruction of property, which may be particularly problematic for people living in rental accommodation. In such cases, these issues may increase risk of relinquishment [7].

Prevalence estimates for separation-related problems vary depending on the population studied. In some questionnaire studies of pet dog owners, prevalence estimates range from 17.2% of dogs [8] to just 5% [9], whilst in dogs presented to behaviour clinics as many as 20–40% are diagnosed with “separation anxiety” [10]. However, viewed alongside findings that up to 50–56% of the pet dog population may display clinical signs of SRBs at some stage in their lifetime [11], it is estimated that up to 70 million dogs across the United States and Europe could present with SRBs during their lifetimes [12]. Validated advice programmes for dog owners that are designed to reduce the occurrence or development of SRBs have previously been established [13,14], and yet research suggests poor owner adherence in implementing these beneficial programmes [13,15]. Whilst the precise reasons for poor adherence to advice for preventing or reducing SRBs have not been documented, it is at least partly attributed to the relative difficulty of adhering to specific aspects of these programmes, with the easier pieces of advice followed more often than the parts that take more commitment and consideration [13].

Recent events relating to the global COVID-19 pandemic have resulted in many dog owners having changed routines. Pandemic restrictions have led to owners spending more time at home and dogs being left alone significantly less often, and for shorter periods than they were prior to the pandemic [16,17]. While at first glance this appears to be advantageous to dogs that may normally have limited company or display SRBs, in most cases, their human household will inevitably have to leave them again when restrictions to curb the spread of the virus lift. Research papers and reports have forewarned that these changes in owner routines back to “normality” could become disadvantageous to dogs by increasing their potential risk of developing SRBs (e.g., [16,17,18,19]). Dog owners have also expressed concerns about the impact that the change in time their dogs are left alone for could have on their dogs’ behaviour when they need to return to their normal routines again [16,20,21,22]. This situation presents a unique opportunity to learn whether sudden changes in leaving hours do indeed increase the risk of dogs developing SRBs, as many have predicted.

Several studies have explored how social restrictions to control the spread of the virus SARS-CoV-2, which causes the disease known as COVID-19, have impacted the behaviour of companion dogs. Bowen et al. [16] surveyed 794 adult Spanish dog owners about the impacts of ‘lockdown’ on their pet, approximately 3.2 weeks after confinement to their homes. Although relatively early into the change of daily routine, 11.8% of owners were already noticing problems with leaving their dogs at home alone, while 39% were concerned their dog would struggle to adapt to the situation once confinement ended. General increases in dogs’ vocalizations (+24.7%) and fear of loud or unexpected noises (+16.9%) suggested that many dogs were experiencing increased stress and frustration during this period. Bowen et al.’s, [16] results also highlighted that 28.5% of dogs were already reported to have a problem with being left alone before the lockdown, although this is a much higher prevalence than would be expected based upon other studies, e.g., [8], which highlights that there may have been a bias in the population studied. Results from various surveys [17,20,21,22] shared similar findings. In the UK, Christley et al., surveyed 6004 adult dog owners and found that many dogs were being left alone less frequently and for less time during lockdown, alongside other changes in their dog’s management [17]. Meanwhile, qualitative analysis of the survey’s results concluded that while owners understood spending more time with their dogs could lead to dogs struggling in the future when left alone, few owners were preparing for this by providing “home alone” training during the restrictions [20] (corresponding to other findings, which have concluded that owner compliance in carrying out beneficial training to support SRBs was often poor [13,15]). Additionally, Shoesmith et al. in the UK, and Bussolari et al. in the US reported the most frequently cited animal guardianship concerns during the lockdown were the possibility of separation-related problems developing once owners returned to work after an extended period at home [21,22], and reduced opportunity to work from home [20,21,22]. These survey data highlight that while bonds between dogs and their owners were being strengthened during this restrictive period, through increased quality time spent together (e.g., increased playtime [20]), increased risk of vulnerability to SRB was predicted. When taken together, these studies clearly indicate a great concern for the impact of SRBs across European and North American dog-owning cultures as a result of pandemic restrictions. The large-scale change in dog-owner routines as a result of the pandemic provides an opportunity to study the impacts of routine changes on SRBs in the pet dog population, acting in essence as a natural intervention.

The main aim of this study was to test the hypothesis that dogs whose leaving hours reduced most (compared to a February 2020 pre-COVID baseline) would be at greatest risk of developing new signs of SRB when leaving hours increased again. To achieve this aim, we utilised survey data gathered from dog owners between 4 May and 3 July 2020 during the first COVID-19 ‘lockdown’ period in the United Kingdom (UK), and a follow-up survey of the same dog-owner cohort completed between 10 October and 2 November 2020, when restrictions had eased considerably.

## 2. Materials and Methods

This project was reviewed and approved by the Dogs Trust Ethical Review Board (ref: ERB036). Data prior to and during the UK’s first COVID-19 lockdown were collected via an online survey of UK dog owners that was open between 4 May and 3 July 2020. At the start of this period (between 4 to 12 of May) the UK was under its strictest ‘lockdown’ restrictions which began on 23 March, with people only allowed to leave homes for limited reasons, including shopping for food, exercise once per day, medical need, and travelling for work when absolutely necessary. The survey and its methods have been previously described [17]. The current paper focuses on survey data regarding leaving patterns and separation-related behaviour observed by the dog owners. Participating dog owners were asked for consent to be contacted for a follow-up survey. Those who consented (*n* = 4670) were then emailed with an invitation to complete the follow-up survey between 10 October and 2 November 2020, when restrictions had been somewhat eased, and many businesses were open for in-person work again, meaning that more dogs would be being left alone again (see Appendix A for further details on this time period).

### 2.1. Study Subjects

A total of 2425 people completed the follow-up survey for the same dog they scored in the original survey (a 51.9% response rate), of which 2285 had previously provided data to indicate whether their dog was left alone or not in February 2020. Of these, 15.3% dogs (*n* = 305) were not left alone at all in an average week in February, and as such, these dogs did not have baseline data for their behaviour when left, so were excluded from further analysis. After additionally excluding dogs aged under 12 months of age, a total of 1807 dogs had baseline data from February, during the first 2020 ‘lockdown’, and data from the follow-up survey in October 2020. All the results presented in this manuscript are for this cohort of dogs, or subsets of this cohort. Of the 1807 dogs in the cohort, 52.5% were female, and 86.1% were neutered. The majority (56.0%) were of a specific breed, whilst 27.4% were a mix of two known breeds, and the remaining 16.6% were of unknown or multiple mixed breed heritage. The mean age for dogs in the sample at the time of the May/June lockdown was 4.25 years (minimum 1 year, maximum 9.25 years) and 71.9% of dogs lived in a single-dog household.

### 2.2. The Surveys

The surveys were hosted online using SmartSurvey^TM^ software (SmartSurvey Ltd, Tewkesbury, UK). Questions in the first survey included dog and owner demographic information, owner reports of dog behaviour, and management/environment of the dog. Many questions required owners to describe the dog’s behaviour/management during the last 7 days to limit recall bias and ensure relevance to the time of the survey (i.e., during the first phase of lockdown), although some were based upon recall for the early/mid-February 2020 period to provide a pre-pandemic baseline, before people had started to alter their lifestyle to avoid virus transmission. Most questions were optional. In the follow-up October survey, the majority of questions were repeated, asking the owners to describe their dog’s behaviour/management within the last 7 days. This paper focuses on data from two questions that asked about time the dog was left alone without human company (number of days in the past 7 days/an average week in February that the dog was left for at least 5 min; longest period of time left alone in the past 7 days/an average week in February) and two questions on whether owners had observed any of a pre-defined list of SRBs when their dog was left, or when about to be left (in the past 7 days/an average week in February). All questions used here can be seen in the format they were used in the Appendix A. A period of a least 5 min left alone in any day was asked about, as it was thought this would exclude times when an owner left briefly such as to take out the bin, which may be a different experience to truly leaving, as the owner would be unlikely to follow the same leaving routine (e.g., donning coats, locking doors, picking up handbags/keys, etc.) so the dog may not respond in the same way. The pre-defined SRBs were based upon behaviours shown to be common, non-specific signs of SRB [5]. The options owners were able to choose from were: vocalising (barked/howled/pined/whined/cried); paced around, or turned in circles or chased his/her tail; chewed or destroyed non-food items other than toys; scratched/damaged around the door, skirting boards, windows or entrance to the house; scratched/damaged furniture; urinated and/or defecated inside the home; other behaviours (please specify). For owners who selected ‘other behaviours (please specify)’ a free text box was provided for them to describe what their dog did when about to be left alone, or when left alone.

### 2.3. Quantitative Analysis

Data were stored in Excel and analysed using R v4.0.5 [23] via RStudio^®^ v1.4.1106 [24]. Initial descriptive analysis included the calculation of frequency and percentage statistics for all questions on leaving patterns and SRBs at the three timepoints of interest: pre-lockdown (early/mid-February 2020), during lockdown (“the last 7 days” preceding the day they completed the questionnaire in May–June 2020) and after easing in the follow-up survey (“the last 7 days” in October 2020). Dogs who were not left alone for at least 5 min in an average week in February were excluded from the analysis, as these dogs did not have a baseline SRB status. All data were paired, and dogs were classified as showing signs of SRB (SRB+) at each timepoint if: (1) they had been left alone for at least 5 min at the given timepoint, and (2) if they showed one or more of any of the pre-defined signs of SRB either when left alone, or when about to be left alone. Dogs that had been left alone for at least 5 min, but whose owners reported none of the pre-defined signs of SRB were categorised as being clear of signs of SRB (Clear). Dogs aged under 12 months at the time of the first survey in May/June were excluded from the analysis, to avoid potential confounding related to the age of the dog, as owners are advised to gradually increase the time that puppies are left alone, to a maximum of four hours [25].

Generalised linear models (GLMs) with logit link functions were used to test for associations between potential predictor variables and the binary outcome variable of interest: SRB status in October (dogs that were not left alone in October were classified as ‘missing’ in the data). To ascertain predictors separately for a change in status from Clear in the February baseline to SRB+ in October, and from SRB+ in the February baseline to Clear in October, the dataset was split into two, depending on the dogs’ February status, and models were built separately for each. All analyses included participants who acquired their dog prior to the end of January 2020 and provided data for all three timepoints. For inferential analyses, univariable analysis was conducted first, followed by backwards stepwise elimination for the selection of multivariable models. Variables were included in the initial multivariable models if *p* < 0.05, and excluded if *p* > 0.1 [26]. Variables were sequentially removed from the models, beginning with the variable with the lowest *Z*-value and highest *p*-value. McFadden’s pseudo-R2 was calculated using the ‘pR2′ function of the *pcsl* package [27].

In place of using raw data for the leaving pattern variables (number of days out of seven the dog was left for at least 5 min and the longest time left alone in a 7 day period), difference variables were calculated by assigning ascending numeric values to the answer categories and subtracting the answer provided for February from the answer provided during lockdown in May/June (to show change in leaving category difference between baseline and lockdown), and the answer provided for February from the answer provided during October (to show change in leaving category difference between October and pre-lockdown baseline). These difference variables were evaluated for distribution against the dependent variable first, and values with too few cases (where SRB+ or Clear dogs had fewer than 10 representatives with a given value) were binned into the nearest value. These variables were then used as predictors in the GLMs in place of the original leaving pattern variables.

### 2.4. Qualitative Analysis

In the follow-up survey, for each leaving option (“when left” and “when about to be left”), owners were also asked the following question: “compared with how the dog behaved in this situation in early/mid-February, his/her behaviour when he/she (was left alone, or was about to be left alone, respectively) over the last 7 days has been” with the following answer options: The same as before; Different to before; Not applicable. For people who answered either question as ‘Different to before’, a free text box was provided as they were asked to describe how their dog’s behaviour had changed. The free text was coded by two researchers (KG and LS) upon the agreement of coding rules. Ten percent of the free text coding for whether a SRB was present and the SRB status classification (worsened, improved or undetermined) was coded by both researchers, and Cohen’s Kappa used to assess inter-rater reliability. The free text boxes linked to the ‘other’ answer options in all questionnaires were also coded by KG and LS. Where owners had provided text in the ‘other’ box that explained their dog’s behaviour when left as not being linked to separation (e.g., the dog was elderly and toileted inside the house on puppy pads, both with and without company) these dogs were not classified as displaying an SRB. Where the ‘other’ text contained descriptions of behaviour that were associated with SRBs (based on those established by Horowitz [10] and Overall [5]) these were coded as an SRB. This included destructiveness (e.g., scratching, chewing, displacing items), dogs being described as anxious, upset, or stressed, shaking, signs of drooling/drooling, vocalisations, elimination behaviour, appetite changes, and excessive grooming or self-injury when left. Ambiguous or unclear answers were coded as such and were not classified as an SRB.

## 3. Results

### 3.1. Days Left Alone per Week for at Least 5 Min

In total, 49.3% of the dogs (all of whom were left for at least 5 min during an average week in February), were not left alone at all during the seven days prior to their owners completing the survey during the May/June lockdown period (Figure 1). Although many dogs had resumed being left alone at the time of the October follow-up survey, 17.1% of dogs were not being left alone at all. Before the pandemic began, in February, 46.2% of dogs were left alone for at least 5 min on 5 to 7 days in an average week. However, leaving patterns changed considerably during both other phases in the study period, with only 8.7% of dogs being left alone on 5 to 7 days a week during the lockdown restrictions in May/June, and 23.0% at the time of the October follow-up.

### 3.2. Longest Time Left Alone

For most dogs that continued to be left alone during lockdown restrictions in May/June, the length of leaving hours reduced. Although leaving hours across the cohort had increased in October, they had not returned to pre-pandemic levels. This is most clearly seen when looking at the category for being left between 3 and 6 h, which represented 43.0% of dogs’ longest time left alone in February, but only 8.2% of dogs that were left in May/June and 24.6% in October (Figure 2). Across all time periods, a minority dogs were left for more than 6 h, although this did represent 9.3% of dogs at baseline in February.

### 3.3. SRB Status

A total of 22.1% of dogs in the cohort (*n* = 400) had at least one sign of SRB reported by their owners at baseline for February 2020. Whether dogs showed at least one sign of SRB or were clear of SRB signs varied across the study period (see Figure 3). During the May/June lockdown, dogs that had data for SRB status (*n* = 912), 20.2% (*n* = 184) had at least one SRB reported, and in October (of *n* = 1510 dogs with SRB data), 17.2% (*n* = 260) had at least one SRB reported.

There were 1407 dogs who were clear of SRB at baseline in February 2020. When looking at SRB status in October, of 1187 dogs who were left alone in October, 117 (9.9%) were reported to have shown at least one SRB.

There were 400 dogs who were reported to show at least one SRB in February. In October, 323 of these dogs were left alone, and 180 (55.7%) were reported to have no signs of SRBs.

### 3.4. Factors Associated with Development of SRB Signs

Data from the 1407 dogs who were clear of SRB signs at baseline in February formed the basis of this analysis. The strongest univariable predictor of SRB status in October for this cohort of dogs was SRB status during lockdown (May/June), with dogs that showed at least one SRB sign in lockdown having 5.38× times the odds of having at least one sign of SRB in October (Table 1). Dog age was also associated with SRB status in October, with older dogs having increased odds of becoming SRB+. The final variable associated with October SRB status was the difference in the number of days dogs were left alone for between the February baseline and lockdown. Here, the smaller the change in the number of days the dogs were left alone, the lower the odds were of a dog showing SRB in October (odds ratio = 0.81). Data for this variable were treated as continuous, and ranged from 0 to −5, with 0 indicating no change, and more negative values indicating greater reductions in the number of days left alone per week (the original range was from 0–7 days, but due to low numbers for some responses, binning of some responses decreased the range). Interaction terms were investigated, but due to very low numbers in some groupings, meaningful comparisons could not be made; hence, interactions terms are not reported.

Following model selection, the final multivariable model contained two variables: SRB status during lockdown, and the change in days left alone during lockdown compared to February (Table 2). Although statistically significant, the Pseduo-R^2^ was small at 4.23%. Dogs who were classified as SRB+ during lockdown had 4.97× greater odds of being classified as SRB+ in October. Furthermore, dogs whose days left alone per week reduced the least during lockdown compared to their February baseline had reduced odds of developing new signs of SRB in October.

### 3.5. Factors Associated with Resolution of SRB Signs

There were 400 dogs who were SRB+ at baseline in February who were included in this analysis. SRB status during lockdown (May/June) was significantly associated with SRB status in October, with dogs that showed at least one SRB sign in lockdown having 3.44 times the odds of having at least one sign of SRB in October (Table 3). The difference in the longest time dogs were left alone between lockdown and February was on the threshold for statistical significance (*p* = 0.051), with reduced odds of dogs having SRB signs in October for those whose leaving hours changed least during lockdown. The median category change in leaving hours for dogs that still had signs of SRB in October was −2 (indicating a two-category reduction in longest time left alone) but for dogs that changed status to Clear in October it was −1 (indicating a one-category reduction in longest time left alone). No other variables were significantly associated with October SRB status in this cohort of dogs and a multivariable model could not be built.

### 3.6. Qualitative Analysis

#### 3.6.1. Inter-Rater Coding Agreement

Percentage agreement of >90% was determined for both whether an SRB was present (coded Yes/No) and whether SRB had worsened, improved or could not be deduced. Chance adjusted agreement was substantial for the former (Cohen’s Kappa = 0.75) and excellent for the latter (Cohen’s Kappa = 0.92).

#### 3.6.2. How Dogs’ SRB had Changed

For the question in the October follow-up survey regarding whether the dog’s SRBs had changed or stayed the same since February, of those owners noting that their dog’s behaviours were ‘different to before’, a majority of 42% (i.e., 54 out of 128 owners) indicated SRBs had worsened when their dogs were about to be left. Similarly, 40% of owners (47 owners out of 117) indicated that SRBs had worsened when dogs were left. However, 31% (40 owners) and 37% (43 owners) of owners indicated that SRBs were ‘different to before’ in a positive way, i.e., improving since February when dogs were about to be left, or left, respectively. It was unclear from the text for 23% of answers, in both the leaving and left ‘different to before’ answers (a total of 30 and 37 owners, respectively), whether SRBs had improved or worsened, while 3% of owners (4/128) indicated that dogs’ SRBs when about to be left had ‘worsened then improved’ in the time since the last survey.

##### 3.6.3. ‘Other’ Separation-Linked Behaviour

Some behaviours described by owners were not classical indicators of SRB; instead, these answers provided an insight into what behaviours owners were perceiving as being linked to separation in their dogs, so are reported here for interest. Dogs watching out of the window while owners departed was one reported behaviour. Other behaviours described included dogs wanting to go with owners when they left, for example trying to “squeeze out” through the door with them. Owners also noted dogs as being “more excited” in greeting them upon their return (e.g., “Since I’ve been working at home, he started making more of a fuss on my return if I’m out for more than a short while”). Some owners cited dogs as being clingier and following them around more at home than they did in February, while other dogs appeared more reluctant to perform leaving routines such as going to the kitchen or getting in their crate—common spaces to leave dogs while owners are out of the house (e.g., “She used to come to the kitchen when we asked her to”).

Some of the other behavioural descriptions provided were more ambiguous or less common. For example, “He seems more aware that I’m going to go out and behaves in ways that stop me”. Complexities in response were also apparent (e.g., “Behaviour more difficult now if one of us goes out, still very good if it is both of us”), as were miscellaneous behaviours that were not considered to represent an SRB issue. For example, “Doesn’t always bring a toy with her when greeting me on return”, which may have many differing reasons for this behaviour change.

## 4. Discussion

The main aim of this study was to test the hypothesis that a large and sudden reduction in time left alone would increase the risk of dogs developing SRBs when left alone again. The restrictions of movement implemented during the initial ‘lockdown’ in Spring 2020 to curb the spread of SARS-CoV-2 served as a form of spontaneous intervention experiment, around which this hypothesis could be tested. To test this hypothesis, we utilised data collected from two surveys of a cohort of UK dog owners, which gathered data on the time the dogs were left alone before COVID-19 restrictions began, during the May/June lockdown, and in October 2020 when virus control restrictions were less severe.

The associations found here between SRBs and changes in leaving patterns support our study’s hypothesis that dogs without pre-existing SRBs whose leaving routines changed most would be at greater risk of developing SRBs following the lifting of pandemic restrictions. One previous study showed that dogs with SRB were more likely to have first developed it around a time of change in their household, compared to dogs who developed other behaviour problems [28]. However, the changes documented were highly varied and whilst some could be linked to changes in time alone (e.g., changes in work routine) not all were obviously so (e.g., recent house moves, introduction or loss of a new dog or human to the household, reductions in exercise received). To our knowledge, the present study is the first empirical evidence to show a direct link between changes in dogs’ time left alone and risk of SRBs. However, the pseudo-R^2^ for the multivariable model predicting development of new SRB signs was small, only 4.23%, meaning that other factors unmeasured in this study are likely playing a large role in risk of SRB development. It is anticipated that a significant factor in the development of new SRB signs for dogs over the course of the pandemic will be dog owner’s behaviour regarding steps taken to mitigate SRBs, such as slowly building up the dogs’ leaving hours again before the need arose to leave them for longer time periods. A limitation of the current study is that we do not know which owners took steps to mitigate against SRBs, nor the exact manner of how these owners introduced their dogs to being left again after restrictions eased.

After accounting for dogs that were not being left alone in October, we found that 1 in 10 (9.9%) of the dogs in this study that were clear of SRBs in February were newly showing one or more SRBs in October 2020. This value is likely to be an underestimate, as are all of the prevalence statistics presented here for the presence of SRB [28], as by its very nature the owners are separated from their dogs when the behaviour is displayed, so many dogs’ SRBs may not be observed. Thus, not all dog owners may be aware that their dogs are displaying SRBs, unless they leave obvious markers of the behaviour such as when dogs toilet in the house or destroy objects or furniture, which are more likely to be detected, and also more likely to be considered problematic to the owner [29]. Behaviour such as vocalising (barking/howling/pining/whining/crying) or pacing, are likely to be missed unless dog owners receive reports from neighbours or can utilise recording or live streaming equipment to remotely check their dog’s behaviour when left. Unfortunately, we did not ask how many owners used such equipment. Whether or not problematic or noticeable for owners, all forms of SRB can indicate that a dog is experiencing an emotional challenge that can be detrimental for their wellbeing. Mendl et al. [30] indicated that SRBs could be associated with a pessimistic cognitive bias in pet dogs, suggesting that they can negatively impact dogs’ mood and emotional predisposition (affective state), causing an otherwise avoidable welfare concern.

Age has been shown to be a significant factor in risk of SRBs in previous research, e.g., [31], with older dogs less able to cope with change or emotional stressors [32]. Here, in the univariable analysis, we found further support for this association with increased odds of developing new signs of SRB for each year increase in dog age. Dogs with new signs of SRB in October had a median age of 4.9 years, compared to dogs that remained clear of SRB who had a median age of 3.9 years. However, the dogs in this cohort were relatively young, with a maximum age of 9.25 years, and this association did not remain significant in the multivariate model. There was no association between age and October SRB status for the dogs who were SRB+ at baseline.

Sex and neuter status have been shown in some studies to be associated with risk of SRB in dogs, although the relationship is not straightforward. A number of studies have shown neutered dogs to be more at risk of SRB [11,28,33,34], and for male dogs to be at greater risk than females [11,33,34], often with interactions between sex and neuter status, e.g., [34]. However, no associations with sex were found in [28], and in [35] intact dogs were shown to be at greater risk. Here, no associations were found between SRB and either sex or neuter status for predicting SRB in the October follow-up. This may be due to the multi-breed nature of the cohort examined here, as sex effects on dog behaviour can be breed specific, e.g., [36], or due to the different methods of measuring SRB, which was only considered as present or absent here, as opposed to measuring severity. As with previous research [37,38,39,40], this study found no association between risk of SRB for dogs that lived as single dogs, or in multi-dog households where they are left alone with the company of other dogs.

When looking at the ‘other’ separation-related behaviours described by dog owners and changes in their dogs’ SRB, attempting to go out of the door with their owners and increased excitement upon return were most often mentioned. Whilst neither of these behaviours are necessarily linked with clinical SRB problems, it may be that dog owners perceive them as being a sign that their dog was unhappy when left, which could lead to feelings of guilt. High levels of excitement in greeting have been shown not to be predictive of SRB [41]; however, as with most dog behaviour research, the study that showed this was cross-sectional. Cross-sectional studies can only capture a snapshot of each dog as a one-off, which by their nature neglects to measure changes over time. It is plausible that changes in a dog’s greeting behaviour over time, as mentioned by owners in this study, could provide more insight into factors influencing the dog–owner relationship. In this case, changes in greeting behaviour could be an early indicator of anxiety about separation, which warrants further investigation. It is possible that there are behavioural precursors to SRB that could act as early warning signs for dogs who may be at risk of developing it. However, without longitudinal studies of separation responses in dogs, this will remain unexplored. Alternatively, attempting to follow their owners out of the door could be associated with other factors, such as anticipation of going for a walk. Since we know that owners’ walking patterns in this cohort also changed during the lockdown period [17], the latter explanation seems plausible.

This study has provided a unique and valuable insight into the impact of sudden routine changes on SRB in pet dogs, but as with all research, there are limitations which must be acknowledged. The study participants were a self-selected cohort of owners who were committed enough to the study to complete two 20–40-min questionnaires, five months apart, and the data for the pre-pandemic baseline in February was based on recall of an average week from a few months previous. However, the participants took part in the study to tell us about various factors of their dog’s behaviour and management, and were recruited from the general dog owning population, so it is unlikely that the sample was biased towards owners of dogs with SRB. As the questions on SRB were part of a larger questionnaire, we were unable to gather detail such as severity of SRB, or whether the behavioural signs indicative of SRB occurred when owners were in the house. The data here were limited to simple presence/absence of behaviours indicative of SRB when owners were out of the house, as reported by the dogs’ owners, and does not constitute a clinical diagnosis of SRB problems.

The results of this study show that 9.9% of dogs were displaying new SRBs at the time of the follow-up survey, but it must be remembered that many dogs were still not being left for as long as they were before the pandemic, so that figure is likely to be an underestimate for how many dogs may go on to develop new SRBs when left for longer periods again. The results presented here support the validity of concerns that have been voiced by both dog welfare organisations (e.g., [42]) and dog owners [16,20,21,22] about the potential negative impact that changes in leaving routine would have on SRB in dogs. Without data on the rate of change in SRB signs over time, where people’s leaving hours do not change, it is difficult to know how much of this result is due to the changes in people’s routine or part of the background rate of change in SRB in the population. However, the association found between risk of newly developed SRB and changes to leaving hours supports the role of routine change as a risk factor for the development of SRB, meaning that preventative measures for gradually building dogs’ leaving hours again, after a sudden change, should be beneficial.

Of the dogs that showed signs of SRB pre-pandemic and were left alone at the time of the October 2020 survey, 55.7% had ceased to exhibit signs of SRB. This is a very positive finding, as it highlights the flexibility of this behaviour, supporting the knowledge that it is changeable. Unfortunately, in this study we did not collect data on whether the owners were taking steps to treat their dogs’ SRB, so we cannot know how many of these cases resolved spontaneously, perhaps due to habituation, or because of behaviour modification plans. Regression modelling revealed a small effect of the change in leaving hours during lockdown on the resolution of SRB signs, with dogs whose leaving hours changed least during lockdown being less likely to show SRB in October. Likewise, a small association that was approaching significance (*p* = 0.060) was found with the difference in longest leaving hours between October and lockdown, with dogs whose leaving hours had increased most being more likely to still show SRB in October. We cannot over-interpret these findings however, as both were weak and just outside the threshold for statistical significance. One thing to remember when interpreting these data is that, overall, dogs were not being left alone as often, nor for as long, in October 2020 as they were in February, prior to the pandemic. Overall, the prevalence of SRB decreased in the population over the study period, from 22.1% to 17.2%. It is possible that this was due to the increased time people had been spending at home during the pandemic period. Even for people who were not able to work from home, many were still likely to be spending more time at home due to limited options for social activities outside of the house, in addition to self-imposed limitations on leaving the home, e.g., [43].

The findings of this study have relevance for dog owners and professionals involved in advising them, such as veterinary surgeons and behaviourists. Whilst there was a proportion of dogs whose behavioural signs when left alone remained unchanged despite management changes, this study highlights that for many dogs SRB occurrence can change with circumstance. Both the development of new SRBs and the amelioration of existing signs when dogs develop new expectations about being left alone indicate that such behaviours are not necessarily ‘fixed’, and changes in management can influence occurrence.

Further research is needed to understand whether taking steps to prevent the development of SRBs over the course of the pandemic had any impact in reducing risk, and a final follow-up survey of this cohort is planned for October 2021 to investigate this. It would be of value for future studies to more closely evaluate dog and owner behaviour when dogs are left alone over a longitudinal period, particularly in response to sudden routine changes, such as over extended holiday periods. Additionally, whilst research shows that owner adherence to behaviour modification plans for SRB is poor [13], it is not yet known what proportion of dogs with SRB resolve their behaviour without specific treatment, nor whether implementing parts of a behaviour modification plan (such as those that are easier to adhere to) may be effective for some dogs.

## 5. Conclusions

The dogs in this longitudinal survey study exhibited significant changes in the prevalence of SRB signs across the study period. Following changes to leaving hours due to pandemic restrictions, one in ten dogs (9.9%) developed new signs of SRB by the follow-up survey in October 2020, with dogs whose leaving hours decreased most during lockdown restrictions being at increased risk of developing new signs of SRB. In contrast, 1 in 2 dogs (55.7%) who showed SRB before the pandemic and were being left alone in October 2020 had ceased to show SRB at that time point. Unlike previous studies, no associations were found for likelihood of showing SRB and dogs being male or neutered, but an association was found between dogs’ age and the risk of beginning to show SRB where dogs did not before, with older dogs more at risk. Overall, the prevalence of SRB fell between February and October 2020, from 22.1% to 17.2%, as the time people spent at home with their dogs increased. These findings have implications for our understanding of the etiology of SRB, and further demonstrate a link between changes in the time dogs are left alone and SRB risk.

## Figures and Tables

**Figure 1 animals-12-00482-f001:**
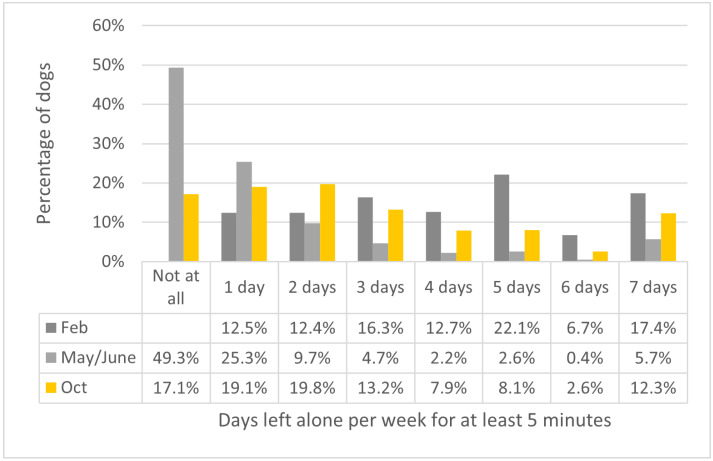
The percentage of dogs in the sample (*n* = 1807) who were left alone for at least 5 min on a given number of days on average at baseline in February, in the previous 7 days for the May/June survey under ‘lockdown’ restrictions, and in the previous 7 days for the October follow-up survey. Dogs who were not left alone at all in February were excluded from this analysis.

**Figure 2 animals-12-00482-f002:**
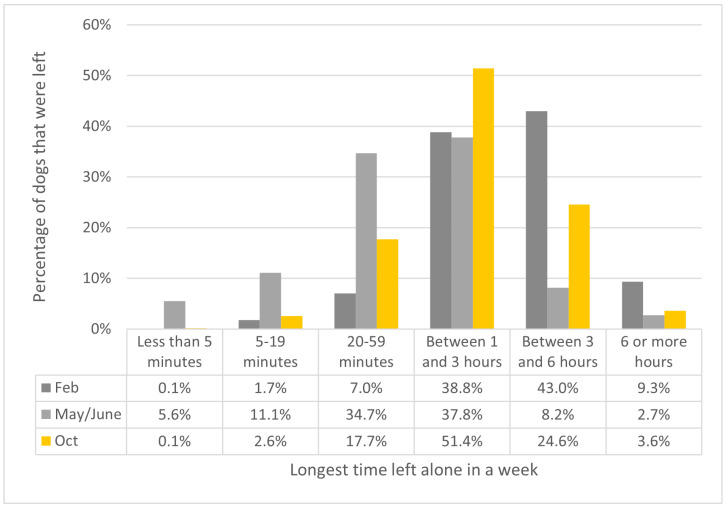
The longest amount of time dogs were left alone, as a percentage of dogs that were left alone in each period.

**Figure 3 animals-12-00482-f003:**
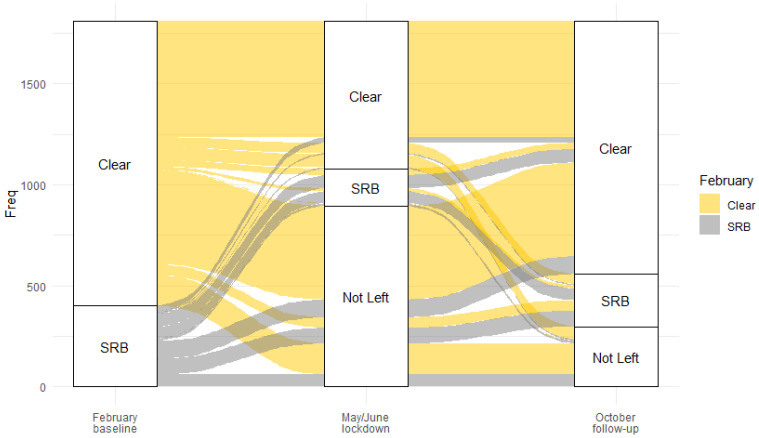
Alluvial (Sankey) plot showing changes in SRB status between the three study timepoints in 2020. The grey bars indicate the changing status of the dogs who were SRB+ in February 2020, and the yellow bars show the dogs were clear of SRB at baseline. The yellow and grey bars split to show movement in the status of those dogs between the other two study points, during the 1st ‘lockdown’, where almost half of dogs were not left alone, through to October 2020 in the follow-up survey.

**Table 1 animals-12-00482-t001:** Results from univariable logistic regression models for *n* = 1407 dogs who were Clear of SRB signs in February 2020. The dependent variable is SRB status in October 2020, of which *n* = 1070 were classified as still Clear in October, but *n* = 117 had transitioned to being SRB+ and *n* = 220 were not being left in October, so had a missing SRB status and were excluded from this analysis. Text in bold highlights variables significant to *p* < 0.05. * Denotes variables treated as categorical. Unless otherwise indicated, variables were treated as continuous.

Variable	Est	Std. Error	Z	*p*-Value	OR
SRB status during lockdown (ref: Clear) *	**1.68**	**0.35**	**4.87**	**<0.001**	**5.38**
Sex (ref: female) *	−0.01	0.20	−0.04	0.972	0.99
Age	**0.12**	**0.04**	**2.86**	**0.004**	**1.13**
Neutered (ref: neutered) *	−0.01	0.30	−0.32	0.749	0.91
Difference: days alone Lockdown–Feb	**−0.21**	**0.06**	**−3.54**	**<0.001**	**0.81**
Difference: days alone Oct–Lockdown	0.01	0.06	1.48	0.140	1.10
Difference: days alone Oct–Feb	−0.07	0.06	−1.19	0.235	0.93
Difference: Longest time alone Lockdown–Feb	−0.05	0.05	−0.95	0.341	0.95
Difference: Longest time alone Oct–Lockdown	−0.03	0.06	−0.48	0.635	0.97
Difference: Longest time alone Oct–Feb (ref: Left alone less) *^					
No change	−0.23	0.20	−1.36	0.174	0.76
Left alone longer	−0.63	0.39	−1.61	0.108	0.53
Single or multi-dog household (ref: single) *	−0.08	0.22	−0.36	0.722	0.92

^ Due to data distribution this variable was split into 3 categories, with the reference category being dogs that were left alone for less time in October than February (i.e., all dogs whose difference value was negative).

**Table 2 animals-12-00482-t002:** Results from the final multivariable logistic regression for *n* = 1407 dogs who were Clear of SRB signs in February 2020. The dependent variable is SRB status in October 2020, of which *n* = 1070 were classified as still Clear (coded as 0) in October, but *n* = 117 had transitioned to being SRB+ (coded as 1) and *n* = 220 were not being left in October so had a missing SRB status and were excluded from this analysis. * Denotes variables treated as categorical. Unless otherwise indicated, variables were treated as continuous.

Variable	Est	Std. Error	Z	*p*-Value	OR
(Intercept)	−2.97	0.23	−13.11	<0.001	-
Difference: days alone per week Lockdown–Feb	−0.22	0.06	−3.45	<0.001	0.80
SRB+ during lockdown (ref: Clear) *	1.60	2.60	3.60	<0.001	4.97

Pseudo-R^2^ = 4.23%.

**Table 3 animals-12-00482-t003:** Results from univariable logistic regression models for *n* = 400 dogs who were SRB+ in February 2020. The dependent variable is SRB status in October 2020, of which *n* = 143 were classified as still SRB+ but *n* = 180 had transitioned to being Clear, and *n* = 77 were not being left so had a missing SRB status. Text in bold highlights variables significant to *p* < 0.05. * Denotes variables treated as categorical. Unless otherwise indicated, variables were treated as continuous.

Variable	Est	Std. Error	Z	*p*-Value	OR
SRB status during lockdown (ref: Clear) *	**1.24**	**0.46**	**2.68**	**0.007**	**3.44**
Sex (ref: female) *	<0.01	0.23	0.02	0.986	1.00
Age	−0.02	0.05	−0.43	0.665	0.98
Neutered (ref: neutered) *	−0.22	0.33	−0.67	0.502	0.80
Difference: days alone Lockdown–Feb	−0.04	0.06	−0.64	0.524	0.97
Difference: days alone Oct–Lockdown	−0.01	0.05	−0.16	0.875	0.99
Difference: days alone Oct–Feb	−0.06	0.06	−0.10	0.319	0.95
Difference: Longest time alone Lockdown–Feb	−0.12	0.06	−1.96	0.051	0.89
Difference: Longest time alone Oct–Lockdown	0.12	0.07	1.88	0.060	1.13
Difference: Longest time alone Oct–Feb	−0.17	0.137	−1.24	0.216	0.84
Single or multi-dog household (ref: single) *	0.07	0.25	0.28	0.777	1.07

## Data Availability

The data presented in this study are available on request from the corresponding authors. The data are not publicly available due to ethical approval of participant informed consent that included survey respondents being informed that we will remove all personally identifiable information before sharing data with Universities and/or research institutions.

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
