# Peer review of "Impact of Changes in Time Left Alone on Separation-Related Behaviour in UK Pet Dogs"

_animals, 2022, doi:10.3390/ani12040482_

Round 1

Reviewer 1 Report

The manuscript: Impact of Changes in Owner Leaving Pattern on Separation- Related Behavior in Pet Dogs in the UK is one more paper on the behavior of dogs in the COVID-19 pandemic developed in the last two years. A catchy topic, but its content  is unoriginal, bringing nothing new to the pre-pandemic science. It seems that the authors of this study have already tackled this topic exhaustively in their previous articles (*Christley, R.M.; Murray, J.K.; Anderson, K.L.; Buckland, E.L.; Casey, R.A.; Harvey, N.D.; Harris, L.; Holland, K.E.; McMillan, 523 K.M.; Mead, R.; et al. Impact of the first COVID-19 lockdown on management of pet dogs in the UK. Animals 2021, 11, 5, 524 doi:10.3390/ani11010005;

*Holland, K.E.; Owczarczak-Garstecka, S.C.; Anderson, K.L.; Casey, R.A.; Christley, R.M.; Harris, L.; McMillan, K.M.; Mead, 529 R.; Murray, J.K.; Samet, L.; et al. “More Attention than Usual”: A Thematic Analysis of Dog Ownership Experiences in the 530 UK during the First COVID-19 Lockdown. Anim. 2021, Vol. 11, Page 240 2021, 11, 240, doi:10.3390/ANI11010240).

The entire first paragraph of Discussion repeats what has already been written in earlier chapters (line 372-379).

Line 382-384   „To our  knowledge, this is the first empirical evidence to show a link between changes in dogs’  leaving routines and risk of SRBs.”

The authors take priority in the empirical demonstration of changes in dogs when the routine of the day is disturbed. And changing the routine of the day is a well-known factor influencing the dog's welfare and behavior, including the occurrence of separation anxiety, so the above-quoted sentence should be deleted because it is untrue. The authors have hardly studied all the publications on how dogs behave when daily routines were changed, so the term "first study" is exaggerated.

Line 444 „This study has provided a unique and valuable insight into…..”  the authors also mark their only unique role in the study, which does not bring new content.

Final decision: reject

Author Response

Our replies to these comments are given in italics below each comment.

The manuscript: Impact of Changes in Owner Leaving Pattern on Separation- Related Behavior in Pet Dogs in the UK is one more paper on the behavior of dogs in the COVID-19 pandemic developed in the last two years. A catchy topic, but its content is unoriginal, bringing nothing new to the pre-pandemic science. It seems that the authors of this study have already tackled this topic exhaustively in their previous articles (*Christley, R.M.; Murray, J.K.; Anderson, K.L.; Buckland, E.L.; Casey, R.A.; Harvey, N.D.; Harris, L.; Holland, K.E.; McMillan, 523 K.M.; Mead, R.; et al. Impact of the first COVID-19 lockdown on management of pet dogs in the UK. Animals 202111, 5, 524 doi:10.3390/ani11010005;

*Holland, K.E.; Owczarczak-Garstecka, S.C.; Anderson, K.L.; Casey, R.A.; Christley, R.M.; Harris, L.; McMillan, K.M.; Mead, 529 R.; Murray, J.K.; Samet, L.; et al. “More Attention than Usual”: A Thematic Analysis of Dog Ownership Experiences in the 530 UK during the First COVID-19 Lockdown. Anim. 2021, Vol. 11, Page 240 202111, 240, doi:10.3390/ANI11010240).

The content of the two papers Reviewer 1 mentions here focussed on the impact of the pandemic on management changes in dogs and on dog owner experiences at a single time point during the most restrictive ‘lockdown’. The current paper is different to either of these, and indeed any other published paper, in that it focusses on changes in separation related behaviour over time, across two different surveys in a longitudinal cohort. The paper is about separation-related behaviour, not about the pandemic. The changes in leaving routine that occurred during the pandemic afforded the opportunity to study what impact this had on a large population of pet dogs over time, that would otherwise have been impossible to capture at such scale. The topic is unique and important in that it adds evidence for an association that has previously been supposed or observed by individual practitioners, but not demonstrated in the published literature, and certainly not on this scale.

The entire first paragraph of Discussion repeats what has already been written in earlier chapters (line 372-379).

Yes, this was purposeful, it provides a summary of the aims & methods to introduce the discussion as not everyone reads papers in a linear fashion. We have not made any changes based on this statement.

Line 382-384   „To our knowledge, this is the first empirical evidence to show a link between changes in dogs’ leaving routines and risk of SRBs.”

The authors take priority in the empirical demonstration of changes in dogs when the routine of the day is disturbed. And changing the routine of the day is a well-known factor influencing the dog's welfare and behavior, including the occurrence of separation anxiety, so the above-quoted sentence should be deleted because it is untrue. The authors have hardly studied all the publications on how dogs behave when daily routines were changed, so the term "first study" is exaggerated.

There is a difference between something that is considered to be “well-known” or folk-knowledge and demonstrable evidence. The sentence quoted here is not untrue, or exaggerated, however, if the Editor wishes us to explore this further, we would kindly request the Reviewer provide us with examples of publications where this has been empirically demonstrated, as we have not found any in our own searches.

Line 444 „This study has provided a unique and valuable insight into…..”  the authors also mark their only unique role in the study, which does not bring new content.

Given the lack of constructive feedback in these statements, no changes have been made based on these comments.

Reviewer 2 Report

This is a well written and clear manuscript which would be of interest to academics, practitioners and dog owners.

Abstract/simple summary: These are clear and well detailed. My only suggestion is that making it clear why February and October were chosen as survey-points may be beneficial. This is discussed later on but highlighting here would be useful.

Introduction: This is clear and in general provides a good background to the topic (bar a couple of points where more expansion would be beneficial).

Aa little more mention of issues of SRBs from owner perspective (e.g. noise complaints and issues with housing; property damage; decline of human-pet bond) could be beneficial (line 56-57). 

It may be helpful to discuss why there is poor owner adherence to programmes to reduce SRB (Line 66-67). 

I wasn't sure if line 79-80 was needed- about dogs having returned to pre-pandemic pattern,  I think it is a little dismissive of your own research which is valuable and applicable beyond the pandemic. 

Line 96-97- some more detail on the findings of these various surveys would be beneficial similarly to Bowen's study, just so provide a clear background/discussion of literature setting scene for your study.

Methods: This is generally well-detailed. 

It may be beneficial to make it clear why cut off of 'at least 5 mins' was used, e.g. justification for this/why this time duration used.

Line 131: There was an error with referencing wanted to flag.

Study periods: It may be useful to spell out a little more here why these survey periods were chosen here, this is covered elsewhere but think could be made clearer here and earlier.

Figure 1- Perhaps just double-check this is referred to in the text. Think either way this figure would benefit a bit of context on why was included/what was showing in context of this study as currently info is a bit generic and didn't totally tie into people's working from home etc.

Results: These were generally well-detailed with clear layout considering scope of results.

However, I don't think the figures were referred to in the text (unless that was the errors highlighted below?)

There were some errors where reference source not found  which would want to look into (line 246; 265; 274; 292; 305; 323 etc).

Line 366-369= I'm not sure it's totally fair to interpret this (the dog not always bringing toy when greeting them) as an anthropomorphic interpretation on the part of the owner- the anthropomorphic interpretation  seems to be from authors of manuscript not owner? They do not seem to state dog is favouring them less unless text was omitted here.

Discussion: The results are appropriately discussed though perhaps more discussion about further study would be beneficial. I felt the discussion about changes in greeting behaviour as an early indicator was interesting and could be expanded.

Some more detail on the application of the findings to practitioners/owners would be beneficial. I do think this is of interest to a range of stakeholders.

Author Response

Thank you for your time in reviewing this manuscript and for your constructive, helpful feedback. We have addressed your comments and responded in the attached file. 

Author Response

(The authors gave the same response as above.)

Reviewer 4 Report

The authors examined how separation-related behaviors in pet dogs changed in relation to time spent away from home by UK owners in February 2020 (pre-pandemic), May-July 2020 (first lockdown during the pandemic), and October-November 2020 (local tier-system for restrictions; follow-up). Their main finding is that close to 10% of dogs developed new signs of separation-related behaviors by the follow-up survey. The sample size is good, and the results are interesting. My specific comments are detailed below.

I found the writing awkward in places. Here are some suggestions that can be made throughout the paper:

Title: change “Leaving Pattern” to “Time Spent Away”; leaving pattern is less clear (it could be frequency of leaving each day)

Line 27: change “reduced” to “decreased”

Line 29: change “dogs whose leaving hours reduce most” to “dogs whose time left alone decreased most”

Line 30: change “leaving hours increase again” to “time left alone increased again”

Line 214: if possible (might not be possible if this was how it was phrased in the survey), change “different to before” to “different from before”

Lines 383-384: Change “dogs’ leaving routines” to “dogs’ time left alone”

Materials and Methods:

Line 151: how was the criterion of “at least 5 minutes” chosen?

My main question when reading this section concerned how owners would know that certain separation-related behaviors occurred when they were away? There would be evidence for some behaviors, such as destructive behavior or defecation, but how would owners know about pacing or tail-chasing? This topic is not addressed until the Discussion section (line 395) but should be addressed here in the Methods and Materials section. Were behaviors, including different vocalizations, defined for owners? For example, what is the difference between pined, whined, and cried?

Also, I might have missed it, but did all of these owners live alone with their dog(s), meaning no other people in the household? If other people were present, did all of them have to be away or just the dog’s owner? This information might be presented in your other paper, but it would be good to include this basic information here as well, so readers can better understand your methods and findings.

Line 166: define “devolved nations” for readers unfamiliar with the term.

Results:

Figure 2: should the blank box for “Feb, Not at all” be 0.0%?

Figure 4 is not called out in the text of the Results section.

Conclusions:

I believe Animals requires a Conclusions section.

Minor issues:

Line 35: insert “of” after 9.9%

Line 100: insert “the” after “in”

An error message occurred in several places (e.g., line 131, line 246, line 265)

Line 148: change “focusses” to “focuses”

Line 355: delete the first “the”

Line 461: change “onto” to “on to”

Author Response

(The authors gave the same response as above.)

Reviewer 5 Report

The study reports that pet dogs whose leaving routines changed most during the COVID-19 pandemic (i.e., their owners left them alone less frequently) would be at greater risk of developing separation-related behaviours following the lifting of pandemic restrictions. The authors gathered data (via owner surveys) about the same dogs before the pandemic, at the beginning of the pandemic and approx. 5 months later. The article is generally well-written, the topic is timely, important, and interesting from a dog welfare aspect. However, currently, the paper seems too long, and it goes into too much detail about how the pandemic took place in the UK. This information would have been interesting if data on the mood, stress level, general mental health of the owners had also been collected in connection with the pandemic and the links with dog behaviour have been investigated. But the questionnaire only asked how long the dogs had been alone and what they were doing during this time. Therefore, the detailed description of the pandemic is unnecessary and could be moved to the Supplemental material, together with Figure 1.

According to the authors, the main finding is that dogs whose leaving time reduced the most during the lockdown were most at risk of developing new SRBs. I think that according to the statistics, the strongest finding is that dogs who showed SRB in February or May had increased odds of having SRB during the subsequent measures which is not surprising. What is surprising though is that 55.7% of the dogs who were reported to show SRB in February (before the pandemic) were clear in October which is good news. However, the authors do not focus on this result and do not explain this finding.

Specific points requiring attention:

L18-20 „Whether dogs showed SRBs or not changed considerably over the months of the study, and one in ten dogs were found to have developed new SRBs in October, that they didn’t show before the pandemic”

Overall, the percentage of dogs showing SRBs decreased during the study, and this should be clarified in the sentence above. What I see in Figure 4 and in the text is that approx. half of the dogs from the SRB group moved to the Clear group in October.

L22 and elsewhere: dogs’ “leaving time” or “leaving hours”

I think this expression is misleading because it is not the dogs that have left home, but the owners. But this is the opinion of a person using English as a second language.

L33 10th instead of “10th”

L45 Separation behaviour, separation-related behaviour, separation distress, separation anxiety terms are used interchangeably throughout the text, although these terms do not necessarily describe the same behaviours. Please provide a clear definition of the behaviour aimed to study and stick to one term.

L92 delete the comma after Bowen et al.’s.

L92-92 28.5% of dogs reported to have SRPs – this is a much higher number than mentioned in previous studies and needs to be discussed.

L118 Materials and Methods

This section (after the Ethics) should begin with the Subjects section. It is confusing to read about the surveys without knowing the sample sizes per survey and the demographics.

L127-137 The dates of data collections have been already mentioned in L121-126, so there is no need for repeating them. The whole section is not relevant to the main question.

L131-132 and elsewhere: (Error! Reference source not found.) This error occurs 7 times in the text, please correct.

L148 Please explain why a periof of 7 days were asked to be reported

L165-167 Figure 1. This Figure should be moved to the Supplement material as it tells nothing about the dogs’ behaviour.

L216-218 “The text was then coded by two researchers”

I could not find agreement data reported later.

L238 Do the authors have information about the Owner demographics? What was the proportion of females? Was there a relationship between age and leaving hours? Or explain why this information has not been used.

L232-242 These sections should be moved to the Materials and Methods as a “Subjects” section

L232 How many owners have filled in the questionnaire altogether? What was the proportion of owners who did not leave their dog at home before the pandemic? Why were they excluded? It is perfectly feasible that some owners did not leave their homes because they were for example, ill, but they did so during the pandemic.

What was the proportion of owners who indicated that they would like to participate in the follow-up study? Among them, what was the proportion of owners who has not responded to the request afterward?

L278-280 “There were 1,407 dogs who were clear of SRB at baseline in February 2020. When looking at SRB status in October, of 1,187 dogs who were left alone in October, 117 (9.9%) were reported to have shown at least one SRB”.

In other words, 1407/1807 (78%) of dogs were clear of SRB in February and 90.1% in October. This is good news, and it is unclear for me why is it reported as bad news. The next section (l281-283) emphasizes that 55.7% of the dogs who were reported to show SRB in February were clear in October. This is a big improvement, I think.

L285 Figure 4. This is a great figure, thank you for including it. However, an explanation would be useful for those readers who are not familiar with this type of depicting.

L291 The odds ratio (OR) is 5.38 and not 4.38 in Table 1.

L293-295 “The final variable associated with October SRB status was the difference in the number of days dogs were left alone for between the February baseline and lockdown”.

OR was 0.81 here. Please explain the relationship between the variables.

L298 Table 1 legend: Add what bold indicates

L307 The odds ratio (OR) is 4.97 in Table 1 and not 3.60

L314 Table 2: bold is not used here although it was used in Table 1

L333 table 3: same: bold is not used here

L382 Please refer to Flannigan and Dodman (2001) here (owner’s work schedule affects separation problems). Flannigan, G., & Dodman, N. H. (2001). Risk factors and behaviors associated with separation anxiety in dogs. Journal of the American Veterinary Medical Association, 219(4), 460-466.

L395-397 “This value is likely to be an underestimate”

The underestimation has already occurred during the baseline.

I think that after the revision, the paper would make an interesting contribution to the journal. I hope this review helps – thank you for the opportunity.

Author Response

(The authors gave the same response as above.)

Round 2

Reviewer 1 Report

As I wrote in the previous manuscript review: “Impact of changes in time left alone on separation-related behaviour in UK pet dogs” is one more paper on the behavior of dogs in the COVID-19 pandemic developed in the last two years. Manuscript has been slightly improved, but my comments were not included. The authors still take priority in the empirical demonstration of changes in dogs when the routine of the day is disturbed.

My final decision: reject
